# Vitelline Warbler (*Setophaga vitellina*) songs, calls, and habitat preferences: novel acoustic descriptions of a range-restricted Caribbean songbird

Wyatt J. Cummings⊙°*, David D. L. Goodman°, Craig D. Layne°, Katherine I. Singer°, M. Whitney Thomas°

Department of Biological Sciences, Dartmouth College, Hanover, New Hampshire, United States of America

° These authors contributed equally to this work.
* Wyatt.Joseph.Cummings@dartmouth.edu

## Abstract

The Vitelline Warbler (*Setophaga vitellina*) is an understudied species endemic to a few small islands in the western Caribbean. Little is known beyond its phylogenetic relationship to other New World warblers. We used island-wide surveys and bio-acoustic recordings to investigate the distribution, vocalizations, and ecology of *S. vitellina* across a significant portion of the species' range on Little Cayman Island. We recorded 417 songs from 91 individuals and analyzed the length, frequency, and shape of various song components. We observed and characterized high variation in the composition and character of songs within the Little Cayman population. We also describe the call of the species and use sound files from across the species' range to compare vocalizations between islands. Vitelline Warbler abundance is highest in dry forest and dry scrub habitats, suggesting that these habitats are most important for the species. Elaboration of the vocalizations of understudied species like the Vitelline Warbler has the potential to further our understanding of avian evolution and behavior. As much still remains to be learned from this species, action must be taken to protect its critical habitats, especially dry forests, among other conservation measures.

## Introduction

Birdsong is a cornerstone of ornithology, with song learning and variation providing key insights into the broader landscape of avian ecology, life history, and behavior [1–4]. In North America, studies of the family Parulidae (wood-warblers) and particularly the genus *Setophaga* have yielded insights about the identification and categorization of songs [5], the effects of abiotic factors on singing behavior [6], geographical gradients of song variation [7], and advances in passive bioacoustic monitoring [8].

**Data availability statement:** All sound files are available from the Environmental Data Initiative (URL: https://doi.org/10.6073/pasta/b74ef130211f8abe0ea20db890ca7f7f). All distribution and abundance data are within the manuscript and its Supporting Information files.

**Funding:** Field work was made possible by the Biology Foreign Studies Program within Frank J. Guarini Institute for International Education at Dartmouth College. Further support was provided by NSF LTER award number 2224545. The funders had no role in study design, data collection and analysis, decision to publish, or preparation of the manuscript.

**Competing interests:** The authors have declared that no competing interests exist.

The Vitelline Warbler (*Setophaga vitellina*), a close relative of the Prairie Warbler (*S. discolor;* Vieillot 1809; [9]), is a poorly studied member of the *Setophaga* genus that could provide insights into Parulid evolution, speciation, and vocalizations. *S. vitellina* is a range-restricted species, found only in the Cayman Islands and the Swan Islands of the southwestern Caribbean. The warbler's small range, currently estimated at less than 135 km$^2$, has resulted in its global classification as "Near Threatened," but little else is known about threats facing the species [10].

Little research has focused on *S. vitellina* beyond its phylogenetics and morphology. Markland and Lovette [11] conducted a thorough phylogenetic investigation of the species, confirming its recent divergence from a common ancestor *S. discolor* and establishing genetic differences between the three subspecies. Of these three subspecies, *S. vitellina nelsoni* Bangs 1919 inhabits the Swan Islands, *S. vitellina vitellina* Cory 1886 is found on Grand Cayman, and *S. vitellina crawfordi* Nicoll 1904 occurs on both Cayman Brac and Little Cayman, the sole island surveyed in this study [12]. Morphologically, the species exhibits sexual dimorphism (Fig 1).

There has been no published research on the singing behavior of *S. vitellina*. Previous to this study, Cornell's Macaulay Library had 25 audio recordings of *S. vitellina* songs [13]. The song has been qualitatively compared to that of Black-throated Blue Warblers (*S. caerulescens* "Gmelin, JF" 1789), which regularly overwinters on Little Cayman [12], but no bioacoustic studies of *S. vitellina* are available to compare its vocalizations to other *Setophaga* species. Songs from *S. caerulescens* and *S. discolor* are presented in S1 Fig as a visual comparison to the songs described in this study. Here we describe the distribution and typology of *S. vitellina* calls and songs on Little Cayman, with the aim of illuminating poorly understood aspects of the endemic songbird's behavior. Additionally, we characterize the population's observed

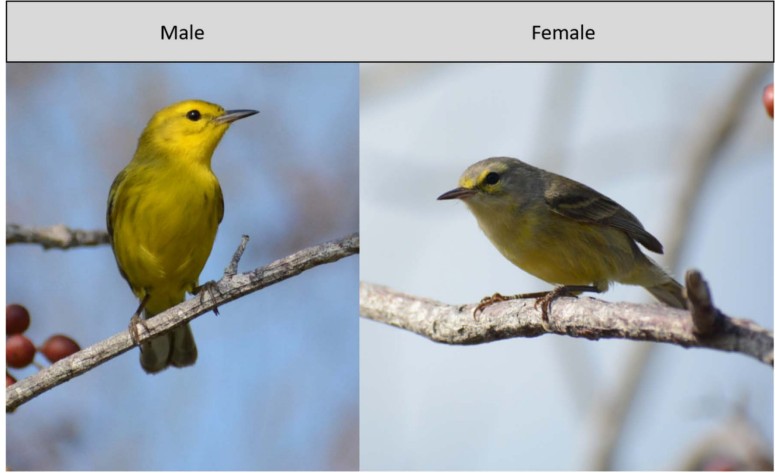

| Male | Female |

**Fig 1. Photographs of each sex of *S. vitellina*.** The male is mostly yellow with darker olive and gray-brown streaks, while the female has a mostly gray head and back, a yellow underside, and olive wings. Photos were taken on Little Cayman on March 6th, 2023 by Wyatt Cummings.

habitat affinities on Little Cayman, offer the first formal description of the species' call, and discuss acoustic differences between the Little Cayman population and individuals from other islands.

## Materials and methods

### Study site

We conducted field research on Little Cayman, Cayman Islands, a low-lying coral island in the western Caribbean. Little Cayman is approximately 16 km long and 1.6 km wide, making it the smallest of the three Cayman Islands. Little Cayman is also the least developed of the islands, with only 161 permanent residents as of 2021 [14]. The vegetation of Little Cayman is dominated by dry forest, dry scrub, sea grape, and mangroves [15].

During this study, we qualitatively categorized portions of the island into these four habitats, supported by our own identifications of the dominant plant species in each. "Dry forest" was defined as vegetation over 2 meters tall, including *Canella winterana* (Pepper Cinnamon), *Coccothrinax proctorii* (Silver Thatch Palm), *Terminalia eriostachya* (Black Mastic), and *Bursera simaruba* (Gumbo Limbo). "Dry scrub," with vegetation under 2 meters tall, featured *Agave caymanensis*, *Zamia integrifolia* (Coontie), *Manilkara zapota* (Sapodilla), *Suriana maritima* (Baycedar), and immature dry forest trees. "Mangrove" habitat occurred in brackish areas and was characterized by the species *Rhizophora mangle* (Red Mangrove), *Avicennia germinans* (Black Mangrove), *Laguncularia racemosa* (White Mangrove), and *Conocarpus erectus* (Buttonwood). "Sea grape" habitat was dominated by *Coccoloba uvifera* (Sea Grape), and occurred almost exclusively within 20 meters of the coast. Some portions of the island, especially along the southwest coast, have more concentrated blocks of developed land.

Little Cayman was a favorable study site for *S. vitellina* because of its low human population density and the proximity of roads to intact forest habitat. The island is circumscribed by a 35.3 km road, with three short connector roads crossing the island, for a total of 40 km of road (Fig 2). This project was completed in cooperation with the Central Caribbean Marine Institute, and all research was done in accordance with the Institute's general research permit, administered by the Cayman Islands Department of the Environment. Additionally, all surveys were conducted on public roads and relied solely on visual and acoustic observations, so no disturbance of Vitelline Warblers or other species occurred.

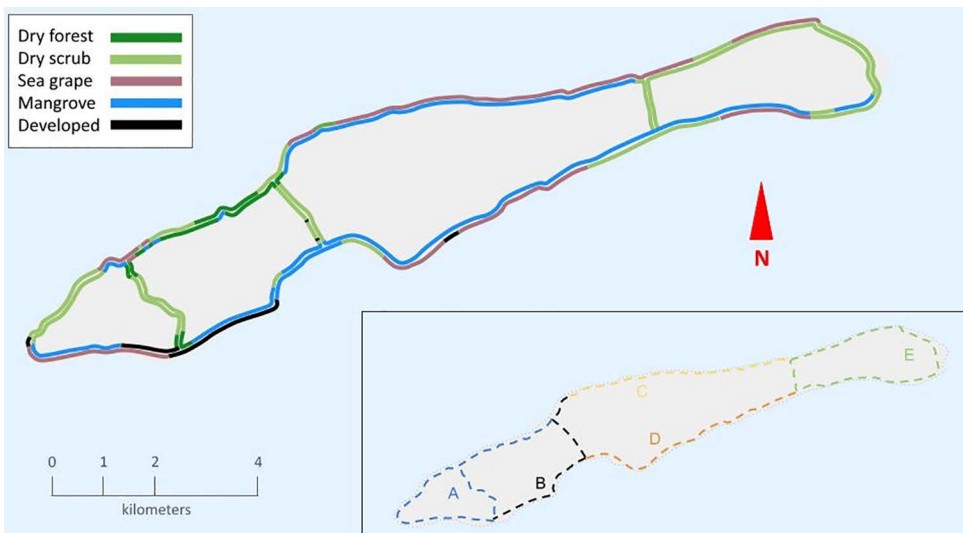

**Fig 2. Map of Little Cayman Island, with colors denoting observed roadside habitat types.** Inset demarcates five distribution survey routes, labeled A-E, which encompass most of the Little Cayman road system. Figure is for illustrative purposes only, with roads and coastlines traced from USGS Earth Resources and Science Center maps (public domain).

## Distributional surveys

We conducted distributional surveys of the entire island between 27 February and 5 March 2023, at the onset of the Vitelline Warbler breeding season. We divided the island's road system into five survey routes of relatively similar length (Fig 2 inset: routes A-E differentiated by color). All surveys were begun in the morning, but as some involved traveling many kilometers farther from the point of origin, the total time of each survey was variable. Survey routes were 10.33 km (route A), 6.27 km (route B), 6.27 km (route C), 6.87 km (route D), and 10.03 km (route E). On each survey, one or more researchers traversed the entirety of the route via bicycle, marking the GPS coordinates of any Vitelline Warbler heard singing using the Cornell Lab of Ornithology's eBird [16] GPS system, which utilizes the network assisted GPS system in smartphones. We surveyed each route once to standardize relative abundance among routes. Surveys were conducted in either sunny or slightly overcast conditions; surveys were postponed in the event of rain or strong winds, as both phenomena may interfere with birds' singing behavior and/or reduce the quality of recordings.

In the field, stretches of roadside habitat were categorized as dry scrub, dry forest, mangrove, sea grape, or developed (houses, lawns, or similarly anthropogenic areas). Later, Google Maps [17] was used to consolidate these observations and estimate the distances surveyed along each habitat block. Because habitat type was often different on each side of the road, habitat blocks were delineated independently on each side of the road (Fig 2). For abundance estimation, the number of birds in a block was divided by half of the length of the block surveyed. This length was halved to signify that one side of the road represents half of the survey route. Information on all habitat blocks surveyed can be found in S1 Dataset.

## Song recordings

We conducted a separate round of recording surveys from 28 February to 8 March, 2023 to collect acoustic data on as many individual warblers as possible. Each singing bird was recorded for three minutes using the inbuilt microphone of a smartphone (Apple and Samsung) and Merlin [18], a smartphone-based identification and recording application. The use of smartphones' inbuilt microphones for bioacoustic studies is uncommon but supported by previous work [19], and allows for greater opportunism and flexibility in sampling. We collected song recordings across the entire breadth of the island in order to account for any geographic differences in singing. All but four recordings were taken in the morning, with the earliest at 6:31 and the latest at 11:44. The four afternoon recordings were taken at 12:02, 14:59, 15:02, and 15:10. Singing was observed from pre-dawn to dusk, but morning sampling was prioritized for consistency. We sought to avoid resampling the same individual, but movement rates of *S. vitellina* over a 24-hour period are unknown, so we cannot rule out the possibility of duplication.

## Data analysis

All analyses of songs and calls were performed with Raven Lite [20]. Analyses were conducted with the following spectrogram specifications for consistency: a frequency range of 0–11 kHz, brightness set at 50, contrast set at 60, and focus set at 690. For figure construction, these settings were modified as needed to yield a clearer image of each song. We visually categorized songs by the number of components they contained (classified as chips, up-notes, and down-notes). Each discrete combination of song components constituted a "song type." For each song, the time and frequency of each component was noted. Time and frequency were measured at the center of the chip's region on the spectrogram, and at the start of the note and the end of the note for up notes and down notes. There were two note varieties (one up-note and one down-note) that we analyzed differently due to their unique shape. The "check" up-note declined in frequency before rising, so we measured time and frequency at the minimum frequency, as well as the start and the end of the up-note. The "parabolic" down-note rose in frequency before declining, so we measured time and frequency at the maximum frequency, as well as the start and the end of the entire down-note.

JMP Pro [21] was used for analyses of time, frequency, shape, and song structure. For each song type, and for all data combined, the Distribution feature was used to assess normality and determine the mean and standard error for each parameter. For comparison of parameter values between song types and between shape varieties of individual song notes, we used a one-way ANOVA test.

Additionally, with permission from the Macaulay Library, we analyzed 29 additional songs recorded and shared by other individuals. Among these are two representative songs each of the Black-throated Blue Warbler (*S. caerulescens*) and Prairie Warbler (*S. discolor*), as well as all 25 available recordings of *S. vitellina*.

### Inclusivity in global research

Additional information regarding the ethical, cultural, and scientific considerations specific to inclusivity in global research is included in the Supporting Information (S1 Checklist).

## Results

### Distribution on Little Cayman

Data for all habitat blocks surveyed on Little Cayman Island, including habitat type, block length, number of singing birds, and latitude/longitude, are provided in S1 Dataset. During our surveys, we encountered 56 singing *S. vitellina*, equivalent to 1.41 singing individuals per km of transect. Distribution surveys suggest clear *S. vitellina* habitat preferences. Abundance was highest in dry forest habitats (4.43 singing birds/km), followed by dry scrub (2.81 singing birds/km), sea grape (0.53 singing birds/km), and mangrove (0.16 singing birds/km). No singing birds were observed in the sections designated "developed" (0.00 singing birds/km). Qualitatively, S. vitellina individuals were often observed foraging in gumbo limbo (*Bursera simaruba*) trees. *Bursera simaruba* were generally infrequent along the coast, and a characteristic tree of elevated, dry areas on Little Cayman.

### Song characteristics

All sound recordings analyzed in this study, as well as accompanying metadata, can be found at the Environmental Data Initiative [22]. The song of *S. vitellina* is composed of three basic components (Fig 3): introductory short notes ("chips"), a long note that rises in frequency ("up-note"), and a long note that decreases in frequency ("down-note").

All analyzed songs (n=417) contained one up-note. The up-note was preceded by 0–4 chips and followed by 0–1 down-notes. These three components—chips, up-note, and down-note—allow for 10 distinct song configurations, which are the only possible combinations of these elements (Table 1, Fig 4). Down-notes were present in 249 songs and absent in 168 songs. Chip notes exhibited variation in structure, length, and frequency. In some songs, chips started at a higher frequency

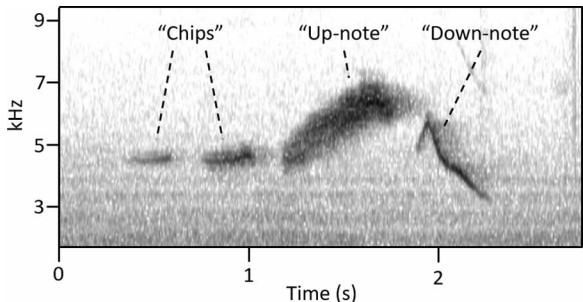

**Fig 3. Spectrogram of a typical vitelline warbler song, with labeled chips, up-note, and down-note.**

**Table 1. All observed song configurations of *S. vitellina*, categorized by the number of chips, up-notes, and down-notes, and ordered by their appearance in Fig 4.**

| Chips (#) | Up-note (#) | Down-note (#) | Occurrence (n) | Fig 4 Identifier. |
|---|---|---|---|---|
| 4 | 1 | 0 | 20 | (a) |
| 4 | 1 | 1 | 9 | (b) |
| 3 | 1 | 0 | 54 | (c) |
| 3 | 1 | 1 | 91 | (d) |
| 2 | 1 | 0 | 61 | (e) |
| 2 | 1 | 1 | 74 | (f) |
| 1 | 1 | 0 | 22 | (g) |
| 1 | 1 | 1 | 57 | (h) |
| 0 | 1 | 0 | 11 | (i) |
| 0 | 1 | 1 | 18 | (j) |

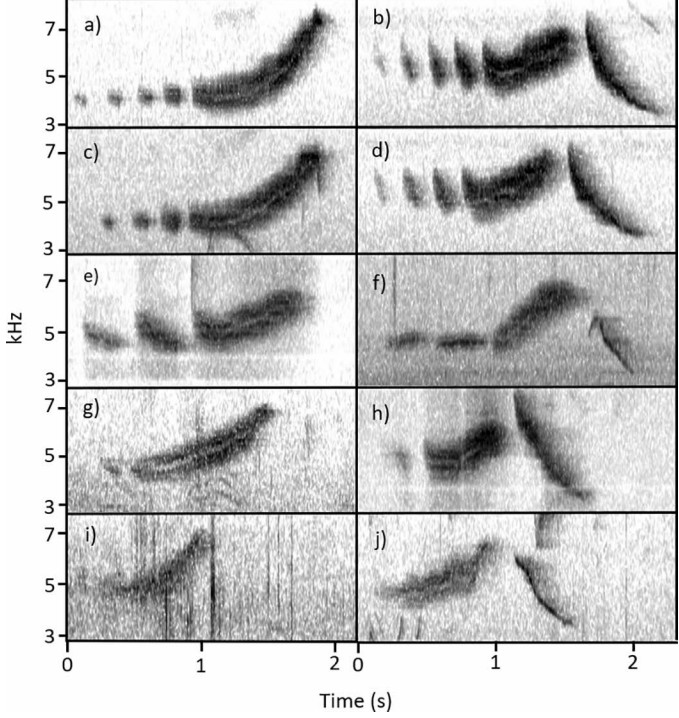

**Fig 4. Ten representative song types of *S. vitellina*, categorized by number of components.** Panels on the left (a, c, e, g, i) lack a down note and panels on the right (b, d, f, h, j) contain a down note. The top two panels (a and b) have four preceding chips, with each successive row below having one fewer chip.

and quickly fell (i.e., Fig 4b; Fig 4d; Fig 4e), while in others, chips were monotone (i.e., Fig 4f). Within individual song components, tone (defined as how much frequency space a vocalization occupies at any point in time) was often variable. Notes that take up less frequency space (i.e., thin high-contrast lines on the spectrogram) sound more like a clear whistle, while notes that take up more frequency space (i.e., diffuse thick lines on the spectrogram) sound more like a buzz.

Individuals were commonly recorded singing more than one song type. Among the 76 individuals that were recorded singing more than once in the three minute period, 24 consistently sang a single song type. The other 52 mostly sang two to four song types (median = 2), but two individuals were recorded that sang six and seven song types.

Across all 10 song types, introductory chips had a mean frequency of 4.79 kHz, up-notes started at a mean frequency of 4.78 kHz and ended at 6.66 kHz, and down notes started at a mean frequency of approximately 6.62 kHz and ended at 3.78 kHz (Table 2).

Up-notes fell into four categories, all increasing in frequency from start to finish. The first shape variety, "check," is characterized by a short initial decline in frequency followed by a longer rise in frequency (Fig 5a). The second variety, "convex," is characterized by an initial steep rise in frequency, followed by a more gradual increase (Fig 5b). The third variety, "concave," is characterized by a consistently low initial frequency that rises exponentially (Fig 5c). The fourth variety, "linear," is characterized by a mostly consistent steepness as it increases steadily in frequency (Fig 5d).

While up-note shapes were categorized qualitatively, quantitative analysis justified our categorization (Table 3). Among the four shapes, initial frequencies differed significantly (One-way ANOVA: $F_{3,416}$ = 97.13, P<0.0001), as did end frequencies (One-way ANOVA: $F_{3,416}$ = 34.98, P<0.0001). The starting frequency of "check" up-notes was higher than that of all

Table 2. Frequency data for all components of 10 Vitelline Warbler song variations.

| Fig 4 ID. | Sample size | Chip frequency (kHz, mean ± SD) | | | | Up-note frequency (kHz, mean ± SD) | | Down-note frequency (kHz, mean ± SD) | |
|---|---|---|---|---|---|---|---|---|---|
| | | 4 | 3 | 2 | 1 | Start | End | Start | End |
| (b) | 9 | 4.99 ± 0.64 | 4.92 ± 0.51 | 4.90 ± 0.49 | 4.87 ± 0.44 | 5.08 ± 0.65 | 6.66 ± 0.39 | 7.23 ± 0.29 | 3.83 ± 0.32 |
| (a) | 20 | 4.64 ± 0.42 | 4.62 ± 0.40 | 4.64 ± 0.35 | 4.65 ± 0.35 | 4.67 ± 0.43 | 6.85 ± 0.38 | | |
| (d) | 91 | | 4.98 ± 0.43 | 4.97 ± 0.40 | 4.97 ± 0.38 | 5.05 ± 0.49 | 6.69 ± 0.38 | 6.79 ± 1.13 | 3.76 ± 0.35 |
| (c) | 54 | | 4.80 ± 0.37 | 4.84 ± 0.36 | 4.84 ± 0.36 | 4.87 ± 0.45 | 6.69 ± 0.56 | | |
| (f) | 76 | | | 4.64 ± 0.30 | 4.63 ± 0.31 | 4.70 ± 0.43 | 6.66 ± 0.46 | 6.56 ± 1.38 | 3.86 ± 0.50 |
| (e) | 59 | | | 4.65 ± 0.31 | 4.65 ± 0.33 | 4.72 ± 0.40 | 6.71 ± 0.52 | | |
| (h) | 57 | | | | 4.59 ± 0.46 | 4.60 ± 0.46 | 6.50 ± 0.54 | 6.29 ± 1.37 | 3.68 ± 0.46 |
| (g) | 22 | | | | 4.60 ± 0.46 | 4.60 ± 0.46 | 6.55 ± 0.44 | | |
| (j) | 18 | | | | | 4.62 ± 0.29 | 6.84 ± 0.57 | 6.83 ± 0.69 | 3.91 ± 0.79 |
| (i) | 11 | | | | | 4.49 ± 0.36 | 6.50 ± 0.48 | | |
| | Mean: | 4.75 ± 0.52 | 4.88 ± 0.43 | 4.78 ± 0.38 | 4.74 ± 0.39 | 4.78 ± 0.48 | 6.66 ± 0.48 | 6.63 ± 1.24 | 3.79 ± 0.47 |

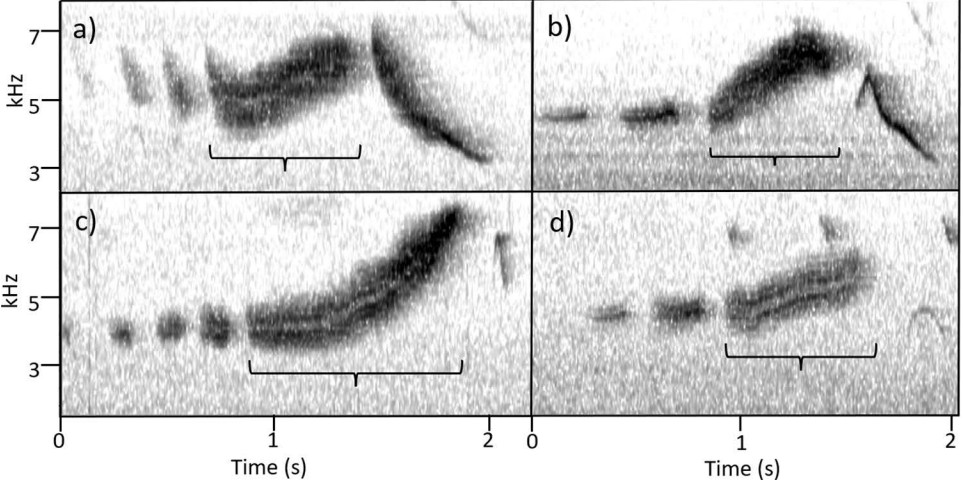

Fig 5. Four *S. vitellina* songs representing the array of up-note shapes. Shapes are labeled: (a) "check", (b) "convex", (c) "concave", and (d) "linear".

**Table 3. Description of the frequency and length of the four up-note variations.**

| | Frequency (kHz, mean ± SD) | | | Component length (s, mean ± SD) | | |
|---|---|---|---|---|---|---|
| Shape | Start | Middle | End | Start-middle | Middle-end | Total |
| Check | 5.38 ± 0.31 | 4.97 ± 0.32 | 6.44 ± 0.43 | 0.12 ± 0.04 | 0.51 ± 0.10 | 0.64 ± 0.11 |
| Concave | 4.62 ± 0.37 | | 6.88 ± 0.38 | | | 0.79 ± 0.15 |
| Convex | 4.51 ± 0.47 | | 6.39 ± 0.54 | | | 0.74 ± 0.16 |
| Linear | 4.71 ± 0.34 | | 6.51 ± 0.49 | | | 0.68 ± 0.17 |

Up-notes of the "check" shape decrease in frequency before rising, so frequency data is represented for the start, low-point ("middle") and end of each note.

three other shapes and the starting frequency of "linear" up-notes was higher than that of "convex" up-notes (Tukey's Honestly Significant Difference Test, P<0.05 for each). The ending frequency of "concave" up-notes was higher than that of all three other shapes (Tukey's Honestly Significant Difference Test, P<0.05 for each).

Down notes fell into five general shape categories, with all ending at a lower frequency than they started (Fig 6). The first variety, "concave," was characterized by an initial steep drop in frequency that attenuates to a near constant frequency. The second variety, "linear," was qualitatively similar to concave, but was characterized by a more consistent decline in frequency. The third variety, "parabolic," was characterized by an initial short rise in frequency, followed by a longer drop in frequency. The fourth variety, "hyperbolic X," was characterized by relatively shallow declines in frequency at the beginning and end, with a steep decline in the middle of the note. The fifth variety, "hyperbolic Y," was characterized by steep declines at the start and end of the note, with a shallower decline in the middle.

In addition to their qualitative differences, down-note shapes differed quantitatively (Table 4). We found significant differences in the starting frequencies (One-way ANOVA: $F_{4,249} = 148.15$, p<0.0001) and end frequencies (One-way ANOVA: $F_{4,249} = 19.96$, p<0.0001) of the five shapes. The starting frequency of "parabolic" down-notes was lower than that of all four other shapes and the starting frequency of "hyperbolic X" down-notes was lower than that of the remaining three shapes (Tukey's Honestly Significant Difference Test, p<0.05 for each). The ending frequency of "parabolic" down-notes was lower than that of all shapes except "hyperbolic X," and the ending frequency of "concave" down-notes was lower than that of both "hyperbolic Y" and "linear" down-notes (Tukey's Honestly Significant Difference Test, p<0.05 for each). Down-notes also appeared to vary considerably in tone. Some notes are clearer (Fig 6c and 6d) while some are more buzz-like (Fig 6b).

The relative occurrence rate of song types among individual birds did not vary with habitat (MANOVA Wilks' λ $F_{30,186} = 1.28$, $p = 0.16$). Recordings of birds singing only once were excluded from the analysis. The sample sizes of songs from mangrove (N=3) and sea grape (N=5) habitats were too small to provide statistical power and the occurrence rates of song types in dry scrub (N=56) and dry forest (N=27) habitats were very similar. There were however notable differences in song attributes among habitats and between shoreline adjacent and inland birds. Birds in sea grape habitat opened their songs with a greater number of chips than the median number for all birds (2 chips) significantly more often than expected randomly, while birds in dry forest habitat opened their songs with two chips more often, and with a greater number of chips less often, than expected randomly ($\chi^2_{df=6} = 36.4$, $p < 0.0001$).

All habitat types, except for sea grape, were found to be located as either shoreline adjacent or inland with dry scrub most readily occurring in either location. Birds located shoreline adjacent opened their songs with more than two chips significantly more often than expected randomly, while birds located inland opened their songs with less than two chips more often than expected randomly ($\chi^2_{df=2} = 42.1$, $p < 0.0001$). Birds located shoreline adjacent also included down notes in their songs significantly more often than expected randomly, while birds located inland also included down notes in their songs less often than expected randomly ($\chi^2_{df=1} = 5.97$, p = 0.015).

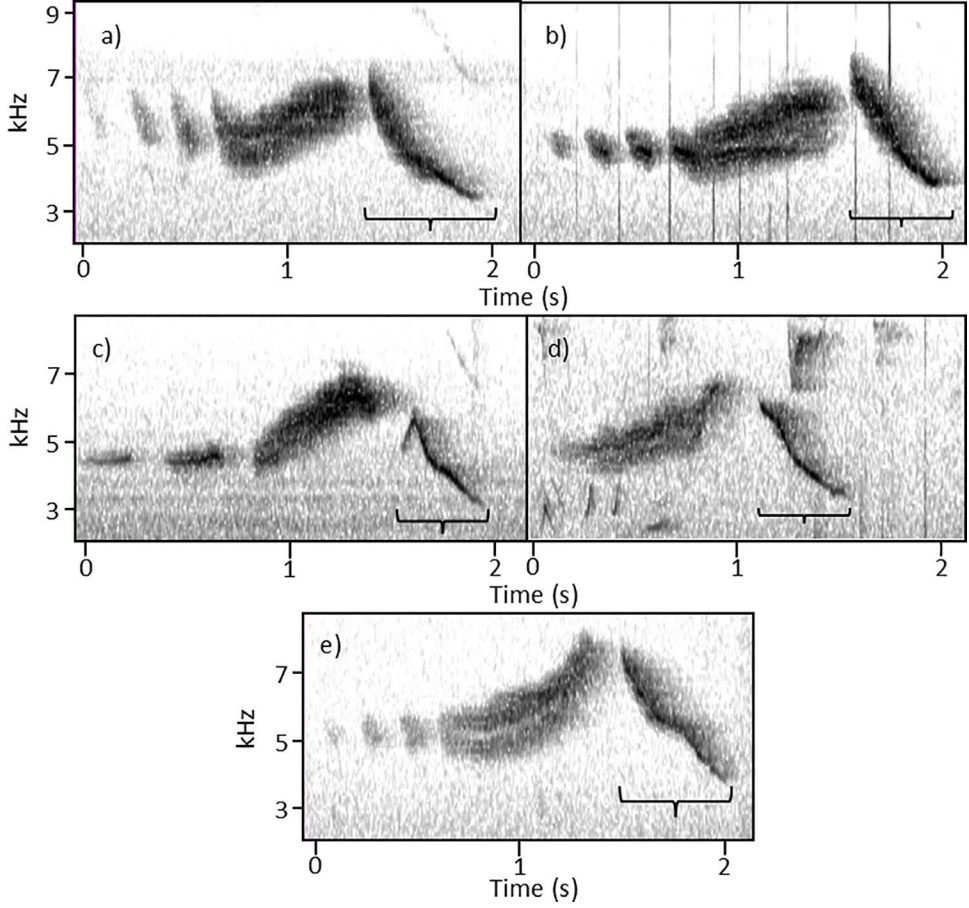

**Fig 6. Five Vitelline Warbler songs representing the array of down-note shapes.** Shapes are labeled "concave" (a), "linear" (b), "parabolic" (c), "hyperbolic X" (d), and "hyperbolic Y" (e).

**Table 4. Description of the frequency and length of five down-note variations.**

| | Frequency (kHz, mean ± SD) | | | Component length (s, mean ± SD) | | |
|---|---|---|---|---|---|---|
| Shape | Start | Middle | End | Start-middle | Middle-end | Total |
| Concave | 7.05 ± 0.70 | | 3.70 ± 0.29 | | | 0.49 ± 0.10 |
| Hyperbolic X | 5.88 ± 0.04 | | 3.68 ± 0.22 | | | 0.33 ± 0.06 |
| Hyperbolic Y | 7.10 ± 0.61 | | 4.10 ± 0.37 | | | 0.42 ± 0.09 |
| Linear | 7.12 ± 0.74 | | 3.99 ± 0.55 | | | 0.41 ± 0.09 |
| Parabolic | 4.19 ± 0.54 | 4.83 ± 0.47 | 3.38 ± 0.45 | 0.06 ± 0.02 | 0.38 ± 0.17 | 0.44 ± 0.18 |

Down-notes of the "parabolic" shape rise in frequency before declining, so frequency data is represented for the start, high-point ("middle") and end of each note.

## Call characteristics of the Vitelline Warbler

There appears to be no published description of the calls of *S. vitellina* [12]. Here, we provide descriptions of calls we observed and recorded.

During one survey, we observed an individual *S. vitellina* calling at close range. Based on the bird's coloration and observed singing behavior, we determined that it was a male. We recorded the warbler's vocalizations for 88 seconds, over the course of which it produced 26 short call notes (three shown in Fig 7). All 26 call notes qualitatively appeared to be the same vocalization. These vocalizations lasted between 0.029 and 0.047 seconds (mean ± SD = 0.037 ± 0.004s). Vocalizations started at a higher frequency (mean ± SD = 6.892 ± 0.094 kHz) and ended at a lower frequency (mean ± SD = 5.811 ± 0.114 kHz). When this same male sang, the songs' chips were at a lower frequency (mean ± SD = 4.97 ± 0.12 kHz), suggesting that calls (not part of a song) and chips (part of a song) are different vocalizations.

## Inter-island variability in Vitelline Warbler songs

The Macaulay Library, a media repository maintained by the Cornell Lab of Ornithology, contained 25 recordings of *S. vitellina* as of February 2025. Each recording contains 1–7 songs, and all were of high enough quality to assess song structure. Of the 25 recordings, 15 were from Grand Cayman, eight were from Little Cayman, and two were from the Swan Islands. Unfortunately, no recordings are available from Cayman Brac, the only other island in the species' range. However, the subspecies found on Cayman Brac (*S. vitellina crawfordi*) is the same found on Little Cayman, so all three subspecies are represented in this set of vocalizations. Full descriptions of these 25 recordings can be found in S2 Dataset. Representative songs from each island (including both available songs that exist from the Swan Islands) are presented in Fig 8.

The categorization scheme for Vitelline Warbler songs on Little Cayman appears to be insufficient to describe variation across the species' entire range. On Grand Cayman, very few songs fell within the variation seen on Little Cayman. Those that matched in terms of overall song structure (5 out of 33 songs across 15 recordings) still differed in the shape and tone of individual song components. For example, when compared to Little Cayman, down-notes on Grand Cayman were often at a lower frequency, exhibited less change in frequency, and often changed in tone from a buzz to a whistle (see Fig 8b vs Fig 8d for a comparison). Other songs on Grand Cayman differed in more extreme ways. Many had 1–3 additional low-frequency notes after the typical down-note (i.e., Fig 8b), and some lacked a distinctive up-note, instead having a continuous series of chip-like notes that rise in frequency (i.e., Fig 8a).

The eight recordings from Little Cayman generally fit with the structural categories laid out in this paper using our own recordings (Fig 8). The songs in these recordings (n=21 across the eight recordings) each had 2–3 chips, always had an up-note, and sometimes (n=8) had a down-note. However, a song in one recording (ML620010641) had an additional note after the down-note which rose in frequency. This type of note was not observed in any of our recordings across Little Cayman but may be a rare variation there.

As only two recordings exist from the Swan Islands, the region's songs likely cannot be fully characterized here. However, the two recordings that do exist both deviate from distribution of vocalizations observed on Little Cayman or Grand Cayman. In one recording (Fig 8e, ML557363371), the four chips are low frequency, in rapid succession, and less buzzy

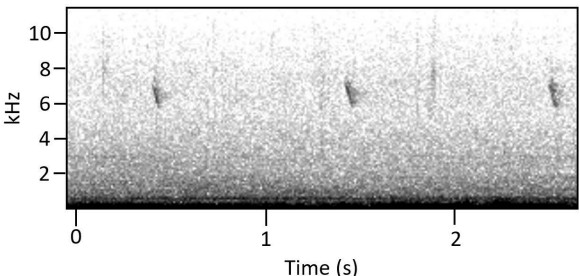

**Fig 7. Call of the Vitelline Warbler.** The species' previously undescribed call is a single clear chip, given at intervals of roughly once per second.

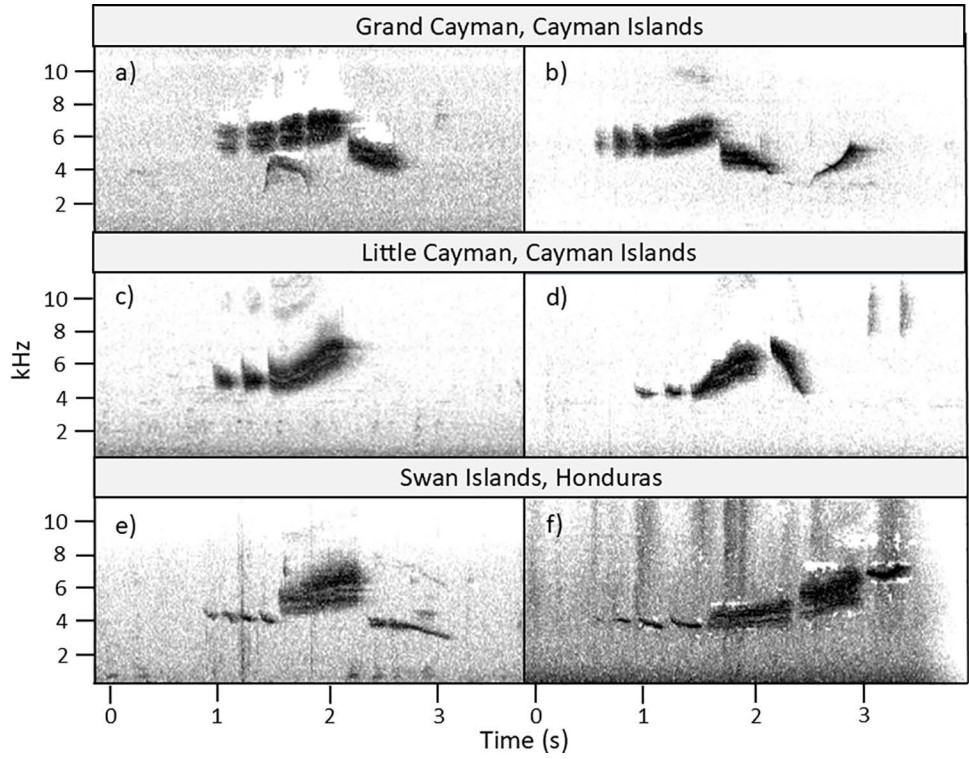

**Fig 8. Six Vitelline Warbler spectrograms from across the species' range.** Songs on Grand Cayman (a-b) and the Swan Islands (e-f) differ in structure from songs on Little Cayman (c-d). Spectrograms were produced from publicly accessible sound files with permission from the Macaulay Library at the Cornell Lab of Ornithology (from a-f, recordings used were: ML148541091, ML28962681, ML620010946, ML 379114681, ML557363371, 451235541).

than is expected, and the down-note is quite long and low frequency. In the other recording (Fig 8f, ML451235541), the traditional up-note is replaced by two long buzzy rising notes, and the final note (which resembles the tone of a traditional down-note) is the highest note in the song.

## Discussion

Our distribution surveys highlight the reliance of Vitelline Warblers on native dry forest and dry scrub habitats. Singing individuals were sparse in sea grape-dominated communities, as well as in mangroves. These results provide a clarified view of the species' habitat affinity.

Our comprehensive song analyses demonstrate variation in both song composition and individual song elements within the population of *S. vitellina* on Little Cayman. Specifically, the existence of the down-note and the observed variety in shapes of up-notes and down-notes represent song structures with no analog in the repertoire of *S. discolor*, by far the closest relative of *S. vitellina*. A visual comparison of the songs of *S. discolor*, *S. vitellina*, and *S. caerulescens* can be found in S1 Fig.

Furthermore, the limited number of song recordings from the rest of the species' range suggest that high variation also exists in other island populations, and songs appear to differ between islands. Song components beyond those observed on Little Cayman were found on Grand Cayman and the Swan Islands, as were new note shapes. While the up-note appears to be ubiquitous in Little Cayman songs, some birds on the other islands had songs with no recognizable up-notes or two up-notes, a striking deviation from the structure observed in our recordings.

The repertoires of song types exhibited by individual Vitelline Warblers on Little Cayman Island are large enough to limit there being any discernible differences in occurrence rate of song types, as we defined them spectrographically, among the habitats. Any variation among habitats may simply not be greater than the variation among individuals within habitats. A greater sample size of songs from the habitats where Vitelline Warblers occur less frequently may challenge that assertion. The greater rates of occurrence for songs with more chips and for songs that include down notes in sea grape habitats, as well as in any habitat that is shoreline adjacent rather than inland, is possibly an example of vocal communication behavior being influenced by ambient background noise. The amount of wind and wave noise drops off quickly away from the beach when the environment contains three-dimensional vegetative structure. But near the beach, chips and down notes can provide greater opportunity for other birds to hear the song.

There are multiple possible mechanisms for the increased variations in song structure of the island-restricted *S. vitellina* as compared to its migratory *S. discolor* relative. High levels of song variation have been observed in other island-restricted songbirds, with proposed mechanisms including reduced selection for song specificity [23] and the non-adaptive accumulation of song errors (deviations from typical songs) that do not greatly influence individual fitness [24]. Furthermore, island bird populations can experience "character release," whereby relatively few competing species allows a trait (in this case song) to become more variable, due to less selective pressure for specialization [25]. However, Morinay et al. [26] compared 49 pairs of island and mainland passerine species and found no general pattern of higher or lower complexity in island species.

It remains unclear whether differences between *S. discolor* and *S. vitellina* are due to evolution of singing behavior in the island species, the migratory species, or both. The last common ancestor of *S. discolor* and *S. vitellina* has been estimated at 1.2 million years ago [11]. The current song of *S. discolor* is relatively simple, composed of many repetitions of the same note type [27], indicating that in the 1.2 million years since their split, either the migratory *S. discolor* song has lost components analogous to the down-note and distinct up-note, or the island-resident *S. vitellina* has evolved these novel song components. Further study of these two warblers, particularly the understudied *S. vitellina,* could help shed light on the mechanisms by which bird song evolves.

## Conservation implications

At least three significant threats face *S. vitellina* today, including feral cats [28,29], climate change, and increasing development on the island [15]. Increased development in the Cayman Islands is particularly concerning, as *S. vitellina* is already classified as Near Threatened due to its restricted range of less than 135 km². Further range contraction could significantly elevate its risk of extirpation and extinction. In light of our findings that the species' abundance varies among four different native vegetation communities, habitat loss will have an outsized effect if dry forest areas are developed. Additionally, conservation efforts are complicated by *S. vitellina's* relative obscurity. Despite the fact that the vast majority of *S. vitellina* are found in the Cayman Islands, they receive little national attention, likely due to their perceived ubiquity and lack of visually striking features compared to the more charismatic Cuban Parrot (*Amazona leucucephala*) [30,31], Red-footed Booby (*Sula sula*) [32], and Brown Booby (*Sula leucogaster*) [12,33].

There appear to be differences between the three subspecies of Vitelline Warbler, at least regarding their vocalizations, but very little is known about the degree to which they differ in their behavior, morphology, and ecological niches. Much remains to be learned about the species' vocalizations, as this study provides a thorough treatment of the Little Cayman population, but not enough recordings exist from other islands for a species-wide summary. Comprehensive study across the species' range is necessary to illuminate such differences, but these disparate populations must be conserved if study is to be possible. The Swan Islands population may be at particular risk, due to its miniscule range and relative isolation, putting it in danger of extinction due to stochastic natural events like hurricanes as well as anthropogenic action.

Vitelline Warblers provide a key case study in divergent song evolution among closely related New World warblers. They should be a conservation priority to preserve the insights they offer into the mechanisms of song evolution in both migratory and non-migratory songbirds. Currently Vitelline Warblers appear abundant and widespread, but increasing

development in the Cayman Islands, climate change, and predation by cats pose future threats, with habitat destruction being the primary concern. To ensure the species' survival, establishing protected areas and controlling development on the island are critical actions. Particular attention should be given to preserving as much dry forest as possible, as well as ensuring this protection happens across the species' range. Furthermore, small islands like Little Cayman offer unique opportunities to address general questions regarding the evolution of birdsong. Preserving these areas is crucial to protect the scientific insights they offer.

## Supporting information

**S1 Fig. Visual comparison of the songs of three *Setophaga* warblers.** Both vitelline warbler songs are taken from our recordings, and the remaining four are used with permission from the Macaulay Library at the Cornell Lab of Ornithology: ML54975241 (Prairie Warbler, left), ML340590901 (Prairie Warbler, right), ML98819 (Black-throated Blue Warbler, left), ML616608500 (Black-throated Blue Warbler, right).
(TIF)

**S1 Checklist. Inclusivity in global research.** Questionnaire detailing permitting and the authors' relationships with the local community.
(PDF)

**S1 Dataset. Distribution and abundance data for Vitelline Warblers on Little Cayman Island.** For each habitat block, spatial and habitat characteristics are reported, as well as the number of singing birds observed.
(XLSX)

**S2 Dataset. Descriptions of all publicly available sound recordings of the Vitelline Warbler.** All recordings are used with permission from the Macaulay Library at the Cornell Lab of Ornithology.
(XLSX)

## Acknowledgements

We thank Victoria Moss for her assistance in conducting distribution surveys and analyzing recordings. We would also like to thank Matthew Ayres, Chris Rimmer, and Miranda Zammarelli for their thorough revisions of draft manuscripts. We thank Pooja Panwar for offering advice on the presentation of results. We thank the Macaulay Library at the Cornell Lab of Ornithology for facilitating access to public sound recordings across the Cayman and Swan Islands. Finally, we would like to thank the staff of the Central Caribbean Marine Institute and all other residents of Little Cayman for welcoming and hosting us for the duration of this project.

## Author contributions

**Conceptualization:** Wyatt Cummings, Craig D. Layne, Katherine I. Singer, M. Whitney Thomas.

**Data curation:** Wyatt Cummings, David D. L. Goodman, Katherine I. Singer, M. Whitney Thomas.

**Formal analysis:** Wyatt Cummings, M. Whitney Thomas.

**Funding acquisition:** Wyatt Cummings.

**Investigation:** Wyatt Cummings, David D. L. Goodman, Craig D. Layne, Katherine I. Singer, M. Whitney Thomas.

**Methodology:** Wyatt Cummings, Craig D. Layne, Katherine I. Singer, M. Whitney Thomas.

**Project administration:** Wyatt Cummings, Craig D. Layne.

**Resources:** Wyatt Cummings, David D. L. Goodman.

**Software:** Wyatt Cummings.

**Supervision:** Wyatt Cummings.

**Validation:** Wyatt Cummings, Katherine I. Singer.

**Visualization:** Wyatt Cummings, Katherine I. Singer.

**Writing – original draft:** Wyatt Cummings, Katherine I. Singer, M. Whitney Thomas.

**Writing – review & editing:** Wyatt Cummings, David D. L. Goodman, Craig D. Layne, Katherine I. Singer.

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
