## [Decision Letter · Decision Letter 0]

7 Jan 2025

PONE-D-24-45551Characteristics of Vitelline Warbler songs and calls: Novel acoustic descriptions of a range-restricted Caribbean songbird (*Setophaga vitellina* )PLOS ONE

Dear Dr. Cummings,

Thank you for submitting your manuscript to PLOS ONE. After careful consideration, we feel that it has merit but does not fully meet PLOS ONE’s publication criteria as it currently stands. Therefore, we invite you to submit a revised version of the manuscript that addresses the points raised during the review process.

We look forward to receiving your revised manuscript.

Kind regards,

Bert Harris

Academic Editor

PLOS ONE

3. Please include a complete copy of PLOS’ questionnaire on inclusivity in global research in your revised manuscript. Our policy for research in this area aims to improve transparency in the reporting of research performed outside of researchers’ own country or community. The policy applies to researchers who have travelled to a different country to conduct research, research with Indigenous populations or their lands, and research on cultural artefacts. The questionnaire can also be requested at the journal’s discretion for any other submissions, even if these conditions are not met.  Please find more information on the policy and a link to download a blank copy of the questionnaire here: https://journals.plos.org/plosone/s/best-practices-in-research-reporting. Please upload a completed version of your questionnaire as Supporting Information when you resubmit your manuscript.

“Field work was made possible by the Biology Foreign Studies Program within Frank J. Guarini Institute for International Education at Dartmouth College. Further support was provided by NSF LTER award number 2224545.”

5. We note that Figure 2 in your submission contain [map/satellite] images which may be copyrighted. All PLOS content is published under the Creative Commons Attribution License (CC BY 4.0), which means that the manuscript, images, and Supporting Information files will be freely available online, and any third party is permitted to access, download, copy, distribute, and use these materials in any way, even commercially, with proper attribution. For these reasons, we cannot publish previously copyrighted maps or satellite images created using proprietary data, such as Google software (Google Maps, Street View, and Earth). For more information, see our copyright guidelines: http://journals.plos.org/plosone/s/licenses-and-copyright.

b. You may seek permission from the original copyright holder of Figure 2 to publish the content specifically under the CC BY 4.0 license. 

Additional Editor Comments:

We have received two very helpful reviews for your manuscript. They both see merit in the paper but they have important suggestions for improvement.

My main concerns are related to the distributional surveys and habitat characterizations. I recognize that these components were not the focus of the paper, but they are critical for conservation.

Title and abstract

Are you sure that the title is appropriate? The acoustics are clearly the focus, but you also collected useful data on distribution and habitat requirements. I think it would probably be better to include something about abundance and habitat in the title. These parts of the paper are important.

Similarly, you include the abundance and habitat methods prominently in the abstract, but you don’t say anything about these results in the abstract. Please revise. Finally, please make the first part of the last sentence of the abstract more specific. What needs to happen to help conserve this species?

Methods and results

More details are required to make the methods replicable.

I agree with reviewer 2 that a clearer statement on data availability is needed. Even if the data will be archived in a public location it would probably be best to have the abundance data and metadata presented, possibly in a summarized form, in the supplementary material. Please specify what time of day you did the surveys by bike, including the earliest time surveys started and the latest time surveys ended. Please also give the length of each transect and the number of times it was surveyed. Was your survey period peak breeding season (so that your detection probability was high)? Under what weather conditions did you do the surveys?

Please specify how you evaluated the habitat and assigned the sampled habitat to a category. Was this done in the field or by satellite images? Did you divide up the transects by habitat? If so, please show this in the data file in the supplementary material and include the GPS tracks or GPS coordinates of the habitat divisions so that future researchers can repeat your surveys and monitor broad changes in vegetation.

I see that the habitat categories are discussed in another reference but please give more details on them. What are the couple most common plant species in each, how tall is the vegetation on average, etc.?

Discussion

Again, please include an interpretation of your habitat and abundance results. Remind us which habitats are most important for the species and give your appraisal of the current status of the species on the island. Also, you list the main threats but what is your opinion about the most urgent threats? And what feasible actions are needed to help the species? Of course more research is needed but please give us your opinion.

Lines 265-266: We need more information on distribution and abundance? Of course it’s always nice to collect more data but you’ve just done pretty thorough surveys of this island. I’m hoping that my requested revisions will clarify how much you learned and make this statement unnecessary.

Delete line 266-267. Instead, give us a clear appraisal of the current status and the necessary actions to help the species.

Reviewers' comments:

Reviewer's Responses to Questions

**Comments to the Author**

1. Is the manuscript technically sound, and do the data support the conclusions?

Reviewer #1: Yes

Reviewer #2: Yes

2. Has the statistical analysis been performed appropriately and rigorously? 

Reviewer #1: Yes

Reviewer #2: Yes

3. Have the authors made all data underlying the findings in their manuscript fully available?

Reviewer #1: Yes

Reviewer #2: Yes

4. Is the manuscript presented in an intelligible fashion and written in standard English?

Reviewer #1: Yes

Reviewer #2: Yes

5. Review Comments to the Author

Reviewer #1: Strengths:

Good writing. Clear and concise.

I like that this study is on a tropical warbler and how the authors compare it to the closely related Setophaga discolor.

The data analysis is simple and easy to understand.

I like the literature review on what is known and what is lacking.

I like the description of density in different habitats.

Good summary of song structure.

Concerns:

The introduction about sub-species in Cayman and Swan Islands gives the impression that this study was conducted in both islands. I would recommend specifying that this study focuses on one of the islands.

There are many tables linked to figures (see below for examples).

We need more information on recording equipment. Were these recorded with phones without external microphones? Recordings with omnidirectional microphones, such as those found in smart phones, do not meet the typical standard for bioacoustics research. Nevertheless, the recording quality is adequate for the purpose of this study.

What time of day did you record birds?

What were the spectrogram setttings?

There seems to be some variation in the tone of the down notes. E.g., the DN’s in Figs. 4f and j appear skinnier on the spectrogram, than those in b, d, can h. I don’t think this difference is an artifact of the spectrogram settings, because the other notes don’t seem to be affected. It would be helpful to interpret this difference.

Chip structure is also quite variable, but that variability is not acknowledged in the text.

Tables 1 and 2: How did you choose these 10 types to highlight? Are they the 10 most common types? The only 10 types? If they are just 10 examples of song types, I would point out that it’s not typical to present representative data like this. While I like how they provide some insight into how songs were scored, they should probably be dropped for concision. Perhaps you could put them in electronic supplementary material. If these are just examples, how many total song types were there?

Did any individuals sing more than one song type?

Details by line number:

Line 50 – 62: I suggest excluding this information since do not interpret songs with respect to bill length or explore differences between males and females.

Line 62: Are they also sexually dimorphic in their vocalizations? Were all songs from adult males?

Lines 72 – 76: I suggest adding that this study intends to compare this species’ vocalizations to those of S. discolor

96. “biked” is colloquial.

Lines 94 - 100: At what times did the authors do the surveys?

Lines 99, 106, 112, 122: Add citations of the programs used?

Line 123: I recommend talking about song types in this species before this sentence.

Line 125: By “shape varieties” you mean the shape of the notes within a song?

Line 221: It would be nice to have spectrograms for the calls as well (after this paragraph).

Line 221 – 228: Any possibility these vocalizations were from females or juveniles? Is the end of February and beginning of March part of their breeding season?

234. Can you cite a description of S. discolor song here?

Lines 263 – 269: I would recommend adding how understanding their song structure is related to conservation implications.

Line 263 – 269: The authors could add future directions in the discussion

Figure 6: Add the bracket that indicates where the down note is in each spectrogram (like in figure 5).

Reviewer #2: This paper provides a report of variation in the songs of Vitelline Warblers on a single island within their range. The data collection and analysis are straightforward, and the study pertains to a poorly known species of some conservation interest. The song typology of the species is relevant to the study of evolution in warblers as the species is a resident, island-dwelling relative of the migratory Prairie Warbler, and has multiple subspecies with poorly defined (perhaps undocumented?) vocal differences. I also like the approach to quantitatively documenting different categories of note variations in the up- and down-notes.

I would like to suggest a few relatively minor additions, which I believe would help increase readership and citations of your paper with little additional effort. The below could be described in statements added to existing sections of the text, or added as their own short sections:

1. Describe how songs compare to those of birds on other islands. If there are no recordings available from other islands/other subspecies, I would note that. From a quick look on eBird, songs differ significantly between islands and subspecies, which may have bearing on taxonomy and conservation of unique populations, and should be included given that the title does not specify that the paper is specific to a certain subpopulation/subspecies.

2. It would be interesting to provide an account of within-individual variation in song type. Do individuals change what song type they use over the course of the 3-min sampling period?

3. Are any song types given at specific times of the day? Some other warbler species have a dawn song.

4. Are there any geographic differences in singing in different habitats/different parts of the island? Assuming not, state somewhere that there appeared to not be geographic or habitat determinants of song type.

Other minor suggestions for edits:

- L49: Paruline  Parulid

- L69-70: Looks like all of the recordings on eBird are of songs, including those not tagged as “song” – 17 recordings in total prior to 2023

- Include a (brief) comparison with Prairie Warbler song in the paragraph at L69-76 in addition to Black-throated Blue

- L233: it would be more accurate to say, “… down-notes represents song components with no analog …” A discussion of complexity would require defining “complexity” in songs and perhaps making a more explicit comparison (because, for example, using a definition that more complex songs have more individual notes, Prairie Warbler songs would be classified as more complex as they can have 25+ notes). Similarly, the following paragraph at L235-245 should be edited for clarity about complexity and what the expectation should be for songs in island birds – more variability in note types? Note number? Frequency range? All of the above? Etc.

- Provide a spectrogram of the short call, as a figure or supplementary figure

- The Conservation Implications section in the Discussion should mention that there exist song differences between the subspecies/different islands

- Include a Data Availability statement in the main text

Overall, the study provides a detailed description of song variability in the Vitelline Warbler, including measures of variability within individual song components. This level of detail is often not documented, especially in species in the tropics. I think the paper would be more well-rounded and have better uptake (e.g., broader readership, more citations) with some changes, detailed above, after which it will be a solid contribution to the literature.

6. PLOS authors have the option to publish the peer review history of their article (what does this mean? ). If published, this will include your full peer review and any attached files.

**Do you want your identity to be public for this peer review?** For information about this choice, including consent withdrawal, please see our Privacy Policy .

Reviewer #1: **Yes: ** David Logue

Reviewer #2: No

---

## [Author Response · Author response to Decision Letter 1]

21 Feb 2025

The following is pasted from our attached document titled "Response to Reviewers":

Dear Bert Harris, reviewers, and editorial team at PLOS One,

I would like to start by expressing how grateful we were to receive your constructive feedback on this paper. We were glad to see your interest in our research, and we agree that your suggestions were all intended to help us improve the impact of this study. I have shared your feedback with the other authors, and we have taken the time to thoughtfully integrate your edits and suggestions. Below, I will detail the changes we made to the manuscript based on your feedback. Attached, you will find an updated manuscript with track changes, an unmarked manuscript without track changes, a new cover letter, and all figures and supplemental files.

In cases where line numbers are referenced, they correspond to the document titled “Revised Manuscript with Track Changes.” Your feedback is presented below in quotes, with our response/explanation presented below each point.

“Please ensure that your manuscript meets PLOS ONE's style requirements, including those for file naming.”

We have revised the manuscript to fit all style requirements of the journal.

“In your Methods section, please provide additional information regarding the permits you obtained for the work. Please ensure you have included the full name of the authority that approved the field site access and, if no permits were required, a brief statement explaining why.”

We have included permit information in both the main text (lines 113-117) and in the questionnaire on inclusivity in global research (S1 Checklist).

“Please include a complete copy of PLOS’ questionnaire on inclusivity in global research in your revised manuscript.”

The checklist has been completed and attached as a supplemental file (SI Checklist).

“Thank you for stating the following financial disclosure: ‘Field work was made possible by the Biology Foreign Studies Program within Frank J. Guarini Institute for International Education at Dartmouth College. Further support was provided by NSF LTER award number 2224545.’

As funders had no role in this study, we have added the following note to a new cover letter, which has been submitted with this manuscript: “The funders had no role in study design, data collection and analysis, decision to publish, or preparation of the manuscript."

“We note that Figure 2 in your submission contain [map/satellite] images which may be copyrighted… If you are unable to obtain permission from the original copyright holder to publish these figures under the CC BY 4.0 license or if the copyright holder’s requirements are incompatible with the CC BY 4.0 license, please either i) remove the figure or ii) supply a replacement figure that complies with the CC BY 4.0 license. Please check copyright information on all replacement figures and update the figure caption with source information. If applicable, please specify in the figure caption text when a figure is similar but not identical to the original image and is therefore for illustrative purposes only.”

Based on your feedback, we have re-created Figure 2 in a way that we feel will be better suited to the narrative of the study while also ensuring proper attribution. Using a public domain (USGS EROS, which we appreciate you suggesting) map of Little Cayman Island, we roughly traced the two features we hoped to represent, the coastline of the island and the main road system. With these features traced, all other manipulations to create the Figure were performed in Google Slides by hand, so no components of the figure should fall under copyright. In the figure caption, we attribute USGS as follows: “Figure is for illustrative purposes only, with roads and coastlines traced from USGS Earth Resources and Science Center maps (public domain).”

“Please review your reference list to ensure that it is complete and correct.”

As requested, we have reviewed our reference list to ensure accuracy and completion. Based on some of the changes made to the manuscript, sources have been added or removed, so the corresponding changes have been made to reference numbers and the reference list.

Additionally, all figures have been run through PACE and have met its standards.

Additional Editor Comments:

“My main concerns are related to the distributional surveys and habitat characterizations. I recognize that these components were not the focus of the paper, but they are critical for conservation.”

We appreciate the interest you have demonstrated in the portion of this project related to distribution and abundance. We agree that this information will have direct relevance for conservation efforts, and may be used for more intentional habitat protection in the Cayman Islands. Therefore, many of our more substantial edits relate to this section of our study. We have clarified the methods related to this section, adding in many more details where useful, and bolstering interpretation of these results in the discussion and abstract.

“Are you sure that the title is appropriate? The acoustics are clearly the focus, but you also collected useful data on distribution and habitat requirements. I think it would probably be better to include something about abundance and habitat in the title. These parts of the paper are important.”

The title has been updated to reflect greater emphasis on our findings related to the species’ habitat preferences and distribution: “Vitelline Warbler songs, calls, and habitat preferences:

novel acoustic descriptions of a range-restricted Caribbean songbird (Setophaga vitellina).”

“Similarly, you include the abundance and habitat methods prominently in the abstract, but you don’t say anything about these results in the abstract. Please revise. Finally, please make the first part of the last sentence of the abstract more specific. What needs to happen to help conserve this species?”

The abstract has been modified to emphasize what was learned from the distribution surveys, provide a clearer statement on conservation needs, and reflect other changes to the manuscript (lines 33-39).

“More details are required to make the methods replicable.”

Overall, we agree with the editor’s comment that more details will make this study replicable. We have made substantial additions to our survey methods, habitat distinctions, and more, which you will see in the manuscript. We hope you will find these changes thorough and appropriate.

“I agree with reviewer 2 that a clearer statement on data availability is needed. Even if the data will be archived in a public location it would probably be best to have the abundance data and metadata presented, possibly in a summarized form, in the supplementary material.”

Based on your comments regarding data availability, we have added new, easier ways to access our data. As a supplemental dataset (S1 Dataset), we have included our habitat/distribution data and metadata. This is referenced in text in addition to the summary information we discuss in prose. Additionally, we cite our EDI dataset in-text so that readers can more easily access sound recordings. In order to give full credit and provide convenient viewing, we have included a summary document (S2 Dataset) describing all sound files received from the Macaulay Library at the Cornell Lab of Ornithology, which are used to make inter-island comparisons.

“Please specify what time of day you did the surveys by bike, including the earliest time surveys started and the latest time surveys ended. Please also give the length of each transect and the number of times it was surveyed. Was your survey period peak breeding season (so that your detection probability was high)? Under what weather conditions did you do the surveys?”

We have expanded the level of detail presented regarding distribution and recording surveys (lines 128-164). Transect lengths, survey times, weather conditions, and associated details are now included and/or expanded. “Peak breeding season” is not documented for the species, but based on casual observations of the birds from other sources and people we spoke with, our window was roughly at the beginning of the observed breeding season.

“Please specify how you evaluated the habitat and assigned the sampled habitat to a category. Was this done in the field or by satellite images? Did you divide up the transects by habitat? If so, please show this in the data file in the supplementary material and include the GPS tracks or GPS coordinates of the habitat divisions so that future researchers can repeat your surveys and monitor broad changes in vegetation.”

A new paragraph (lines 98-109) is added detailing how we assigned habitat categories, and what the dominant plant species are in each. These observations, made by our team while in the field, were thought to be supplementary originally, but we agree that they provide useful context for future researchers, and are now featured in the text. Another paragraph (lines 142-149) describes how we translated these observations into usable data and present these data in the manuscript. Additionally, a supplemental dataset (S2 Dataset) provides comprehensive descriptions of each habitat block, including GPS points and which transect a block belongs to, among other details.

“I see that the habitat categories are discussed in another reference but please give more details on them. What are the couple most common plant species in each, how tall is the vegetation on average, etc.?”

See the new paragraph described above (lines 98-109) for detailed habitat descriptions.

“Again, please include an interpretation of your habitat and abundance results. Remind us which habitats are most important for the species and give your appraisal of the current status of the species on the island. Also, you list the main threats but what is your opinion about the most urgent threats? And what feasible actions are needed to help the species? Of course more research is needed but please give us your opinion.”

The discussion now features direct interpretation of our distribution data, and we connect our finding that abundance is highest in dry forest habitats to a prescription that those are the areas conservation must prioritize. We are clear that habitat destruction poses the greatest danger, and dry forests should be protected. We have augmented the discussion to better tell this story. Overall, the discussion argues two main points: the species has a unique and variable singing behavior, and may offer valuable insights into birdsong evolution; the species relies on dry forest habitat, so greater protection of these habitats (and mitigation of other risks) is necessary. The discussion also now features comparisons among the subspecies/populations and connects these differences to conservation needs.

“Lines 265-266: We need more information on distribution and abundance? Of course it’s always nice to collect more data but you’ve just done pretty thorough surveys of this island. I’m hoping that my requested revisions will clarify how much you learned and make this statement unnecessary.”

We agree that a clarified statement on habitat preferences is more useful here, see above for details.

“Delete line 266-267. Instead, give us a clear appraisal of the current status and the necessary actions to help the species.”

This line has been replaced with: “Currently Vitelline Warblers appear abundant and widespread, but increasing development in the Cayman Islands, climate change, and predation by cats pose future threats, with habitat destruction being the primary concern. To ensure the species' survival, establishing protected areas and controlling development on the island are critical actions. Particular attention should be given to preserving as much dry forest as possible, as well as ensuring this protection happens across the species’ range.”

Reviewer #1:

“The introduction about sub-species in Cayman and Swan Islands gives the impression that this study was conducted in both islands. I would recommend specifying that this study focuses on one of the islands.”

The introduction has now been modified to specify that our focus is on the Little Cayman population. As we have now incorporated an inter-island comparison, we believe discussion of the sub-species and their ranges is still valuable in the introduction.

“There are many tables linked to figures (see below for examples).”

We believe the tables linked to figures provide consistency between visualizations and quantitative characterization of songs and their components. We have kept them as they originally were, but if a different specific method of data presentation would be appropriate/useful, we can certainly augment our figures and tables as needed.

“We need more information on recording equipment. Were these recorded with phones without external microphones? Recordings with omnidirectional microphones, such as those found in smart phones, do not meet the typical standard for bioacoustics research. Nevertheless, the recording quality is adequate for the purpose of this study.”

We clarify in text (lines 153-158) that we did in fact rely on the built-in microphones of smart-phones. We appreciate the additional note that “the recording quality is adequate for the purpose of this study,” and we believe that for the purposes of this primary bioacoustic survey, use of smartphones was appropriate due to the flexibility they provide in opportunistic sampling. In the future, it would be ideal if researchers can follow up on this study by using more sophisticated recording equipment to document these vocalizations. We also attach a new reference where other researchers found the use of smartphones with inbuilt microphones to be useful and beneficial for a bioacoustic study.

“What time of day did you record birds?”

We have expanded the level of detail presented regarding distribution and recording surveys (lines 128-164). Transect lengths, survey times, weather conditions, and associated details are now included and/or expanded.

“What were the spectrogram setttings?”

The methods have been clarified as follows: “Analyses were conducted with the following spectrogram specifications for consistency: a frequency range of 0-11 kHz, brightness set at 50, contrast set at 60, and focus set at 690. For figure construction, these settings were modified as needed to yield a clearer image of each song.”

“There seems to be some variation in the tone of the down notes. E.g., the DN’s in Figs. 4f and j appear skinnier on the spectrogram, than those in b, d, can h. I don’t think this difference is an artifact of the spectrogram settings, because the other notes don’t seem to be affected. It would be helpful to interpret this difference.”

As a full interpretation of tone would require a more in-depth re-analysis of all songs, we add in brief discussions of tone to the results section. See lines 233-236 and lines 304-305.

“Chip structure is also quite variable, but that variability is not acknowledged in the text.”

Information related to chip structure is added to the results: see lines 230-236.

“Tables 1 and 2: How did you choose these 10 types to highlight? Are they the 10 most common types? The only 10 types? If they are just 10 examples of song types, I would point out that it’s not typical to present representative data like this. While I like how they provide some insight into how songs were scored, they should probably be dropped for concision. Perhaps you could put them in electronic supplementary material. If these are just examples, how many total song types were there?”

In text, we clarify that these 10 song types represent all observed combinations of chips, up-notes, and down-notes that we saw in the population. In that sense, we are characterizing all observed variation in the construction of songs from their component parts, not simply 10 possible combinations. With this clarified in the text, we hope that the purpose of the figure

---

## [Editor Report · Decision Letter 1]

17 Mar 2025

Vitelline Warbler songs, calls, and habitat preferences: novel acoustic descriptions of a range-restricted Caribbean songbird (*Setophaga vitellina* )

PONE-D-24-45551R1

Dear Dr. Cummings,

We’re pleased to inform you that your manuscript has been judged scientifically suitable for publication and will be formally accepted for publication once it meets all outstanding technical requirements.

Kind regards,

Bert Harris

Academic Editor

PLOS ONE

Additional Editor Comments (optional):

Thank you for your close attention to our comments. I only have the following small requests:

Title: move the scientific name immediately after the common name.

Line 63: remove period after “study”

Line 76 S. caerulescens and S. discolor should be italicized

Lines 191-192 Bursera simaruba should be italicized

Line 225 S. vitellina should be italicized

Line 370 ‘on Little Cayman’ should not be italicized

Line 386 Capitalized 'Vitelline Warblers'

Line 415 It would be better to not start your conservation implications section by saying ‘more studies are needed’. I would focus on your results and interpretations and then say this at the end if needed (I’m not sure it is).

Line 419 It’s Near Threatened, not Near Vulnerable
---

## [Editor Report · Acceptance letter]

PONE-D-24-45551R1

PLOS ONE

Dear Dr. Cummings,

I'm pleased to inform you that your manuscript has been deemed suitable for publication in PLOS ONE. Congratulations! Your manuscript is now being handed over to our production team.

Kind regards,

on behalf of

Dr. Bert Harris

Academic Editor

PLOS ONE